# Quality Assurance for Hepatitis C Virus Point-of-Care Diagnostics in Sub-Saharan Africa

**DOI:** 10.3390/diagnostics13040684

**Published:** 2023-02-12

**Authors:** Evans Duah, Evans Mantiri Mathebula, Tivani Mashamba-Thompson

**Affiliations:** Faculty of Health Science, School of Health Systems and Public Health, University of Pretoria, Pretoria 0002, South Africa

**Keywords:** REASSURED, evaluation, regulatory standards, HCV, POC testing, SSA

## Abstract

As part of a multinational study to evaluate the Bioline Hepatitis C virus (HCV) point-of-care (POC) testing in sub-Saharan Africa (SSA), this narrative review summarises regulatory standards and quality indicators for validating and approving HCV clinical diagnostics. In addition, this review also provides a summary of their diagnostic evaluations using the REASSURED criteria as the benchmark and its implications on the WHO HCV elimination goals 2030.

## 1. Introduction

Hepatitis C virus (HCV) infection remains one of the world’s most devastating viral infections [1,2,3,4]. HCV infection has varying distribution rates globally, with cases reported in all World Health Organisation (WHO) member regions. HCV infection is commonly recorded in high-risk populations such as men who have sex with men (MSM) [5,6,7], people living with HIV (PLHIV) [4,7,8], persons who inject drugs (PWID) [7], prison inmates [7], pregnant women [4,7], and blood donors [4,7]. According to the WHO 2019 data, over 58 million HCV cases have been reported globally as of 2019, with 1.5 million new cases annually and 290,000 deaths [9]. More specifically, the most recent 2021 data showed that the Eastern Mediterranean and European regions carry the most chronic HCV infection burden, with 12 million reported cases in each region [9]. This is followed by the South-East Asia Region and the Western Pacific Region with 10 million in each region and the Americas with 5 million reported cases [9]. Although the HCV prevalence has been estimated at between 2% and 3% in sub-Saharan Africa (SSA), Central Africa and West Africa regions are disproportionally affected, with mixed modelling estimating HCV prevalence of 7.82% and 4.14%, respectively [4,10,11].

The WHO considers and recommends testing for HCV for high-risk populations, with linkage to treatment and care pivotal in achieving HCV elimination in 2030 [12,13,14]. This is because less than 5% of people living with viral hepatitis globally are aware of their status [15]. This testing requires a two-step HCV diagnosis algorithm, and thus initial point-of-care (POC) testing using anti-HCV antibody testing [12]. This is followed by a confirmatory test for HCV viremia using either a conventional laboratory-led qualitative or quantitative HCV nucleic acid test (NAT) such as RT-PCR or POC HCV RNA assays [12]. Several diagnostic tests are deployed globally to meet the HCV diagnostic demands including rapid tests [16,17].

Unlike high-income countries, resource-limited and hard-to-reach settings, especially in SSA, are faced with challenges such as inaccessible and expensive laboratory-led conventional HCV diagnostics that inconvenience end-users. This promotes health inequalities as a result of poor access to HCV diagnostic services [14,18,19]. POC rapid testing has proven to provide equitable near-site access to diagnostic services for patients, irrespective of their geographical setting and socio-economic status; hence, LMICs largely fall on this diagnostic alternative [20,21,22,23]. This has increased demands in the in vitro diagnostics (IVD) market. Consequently, these demands have increased the number of guidelines and recommendations for quality checks and assessments on IVD test performance [24]. For instance, IVDs are evaluated on their acceptability and usability by target populations (end-users), especially in LMICs. In addition, they are evaluated on the intent for using them aside from the quality markers that influence their approval into the IVD market [25]. This narrative review summarises the regulatory standards and quality indicators for validating and approving clinical diagnostics, particularly in SSA. We focused on real-time connectivity, ease of specimen collection, affordability, sensitivity, specificity, user-friendliness, rapidity and robustness, equipment-free or simplicity, and deliverability to end-users (REASSURED) as criteria for HCV POC diagnostics.

## 2. Global Regulatory Standards for Clinical Diagnostics

In vitro clinical diagnostic devices are used worldwide and are ethically bound to follow international standards [26,27,28]. These standards carefully guide the fundamental principles of quality performance, uniformity, safety, accessibility, and timely delivery of the diagnostic for its intended use. The global standards, as summarised in Table 1, are defined by regulatory bodies such as the WHO, Global Harmonization Task Force (GHTF), International Medical Device Regulators Forum (IMDRF), Asian Harmonization Working Party (AHWP), International Organization for Standardization (ISO), International Electrotechnical Commission (IEC), and Clinical and Laboratory Standards Institute (CLSI) [27,29,30,31,32].

The WHO guidelines are given international preference as they form the basis for most of the national and jurisdictional standards, while the ISO has developed standards for several IVDs [31]. For example, the ISO/AWI 15193/4, ISO 6717:2021, ISO 21151:2020, ISO 17822:2020, and ISO 18113-1:2009 are some of the ISO standards that guide the development and use of medical devices and IVDs. The ISO 22870:2016 spells out the specific quality management systems for evaluating, approving, certifying, purchasing, installing, and maintaining POC test devices [30].

There are also continental standards aligned with international guidelines, as demonstrated in Table 1. These are developed by regulatory bodies within the jurisdiction. For example, the European Union (EU) on 26 May 2022 made a historical transition from a Directive on In Vitro Diagnostic Medical Devices (IVDD) (Directive 98/79/EC) to a new Regulation on In Vitro Diagnostic Medical Devices (IVDR) (Regulation (EU) 2017/746) [33]. The new regulations reaffirm the existing regulations and standards and better align with international standards. In the EU, IVDs are assessed for their intended use and certified following the IVDD/IVDR standards by a regulatory body. However, manufacturers can also self-certify their products [33]. Standardised and self-certified products bear the label “Conformité Européenne” (CE), which indicates that the product meets EU safety, health, and environmental protection, thus being permitted to be distributed across EU borders.

In the UK, IVDs must meet the regulatory requirements of the UK Medical Devices Regulations (UK MDR) 2002 [34]. On the other hand, the Americas’ Food and Drug Administration (FDA) regulates IVDs, subjecting them to scrutiny using rigorous standards aligned with the WHO Prequalification Process (WHO PQ) [35]. The FDA also performs device registration and listing, pre- and post-market controls, and product recalls [36]. Similarly, Asia–Pacific’s digital Health Regulatory body (APACMed Digital Health Committee Regulatory Working Group) harmonises and regulates medical technology and medical devices across the member region [36].

The Acute Care Testing organisation developed a five-stage pocket checklist for evaluating POC diagnostics. This was developed by multi-industry international researchers by engaging in a Delphi process in a series of structured interviews and questionnaire administration [37]. This checklist evaluates POC diagnostics on their clinical pathway, patient stakeholders, economic evidence, test performance, usability, and training.

## 3. Regulatory Standards for Clinical Diagnostics in Sub-Saharan Africa

Unlike the EU, Asia, and most jurisdictions, the sub-Saharan Africa (SSA) sub-region does not have a universal regulatory standards authority for validating diagnostics (Table 1). Instead, countries rely on their in-country standards as stipulated by their Food and Drug Administration (FDA), National Regulatory Authority (NRA), or National Standards Authority (NSA), which wholly or partly conform with WHO and international regulatory standards. However, the functioning and capacity of those country-specific regulatory agencies are limited. More specifically, the WHO has estimated that 90% of 54 NRAs have minimal or no capacity to execute their function [38]. It is therefore difficult to register and move medical products across African borders, unlike in the EU and Asia, without being subjected to country-specific evaluations and certifications. Practically, the product goes through a cumbersome, complex, and sometimes non-transparent system to become evaluated and certified [39]. However, most manufacturers hope to fall on the African Continental Free Trade Area (AfCFTA) to address these challenges [40].

The lack of a continent-wide regulatory standard authority to complement country efforts has posed several challenges in validating the safety, quality, and clinical performance of medical diagnostic products marketed in SSA. A study by Rugera et al., 2014, revealed wide negligence in regulating IVDs in the member states of the East African Community (EAC) [41]. The few countries that regulate IVDs in EAC rely on data generated by research laboratories, which may not account for the variance observed with field-based or population-based evaluations and post-market surveillance.

However, the advent of the WHO PQ process, launched in 2010, has contributed to the strengthening of IVD product access and acceptance in SSA [42]. This process provides valuable product selection and quality assurance services to manufacturers, regulatory bodies, and procurement agencies. In effect, the process meets the missing universal regulatory standard needs of sub-Saharan Africa.

Ghana is a high consumer of IVDs and hence runs a comprehensive assessment of both imported and locally produced medical devices [43,44]. This mandate by law is entrusted to the FDA, an agency under the country’s Ministry of Health [45]. They follow the WHO and ISO standards to regulate medical devices to ascertain their intended use, quality, and safety to ensure public safety (Table 1). However, the FDA does not evaluate medical diagnostics on the fundamentals of accessibility and affordability, which are part of the REASSURED criteria for diagnostic systems. That is a gap left to the discretion of the public and end-users, who may not have the capacity to understand and decide.

## 4. The REASSURED Evaluation

The need for innovative technologies for accurate and timely diagnosis of common diseases in poor settings received a positive response from the clinical research and manufacturing industry. Novel ideas and technologies have been tested in proof-of-concept studies to meet this call of testing at POC. These range from the use of cheaper readily available materials and samples to mobile phone-mounted technologies [46,47,48]. However, these technologies must meet at least a component of accepted standards.

Unlike laboratory-led medical diagnostics, in vitro POC diagnostics must meet the fundamentals for their classification as POC diagnostics. They must be affordable, accessible, and safe, and must meet high accuracy standards. Most laboratory-led tests are met with high-cost implications and may be inaccessible to hard-to-reach poor communities around the world [21].

In 2006, the WHO/TDR made a recommendation for the full implementation of the ASSURED (Affordable, Sensitive, Specific, User-friendly, Rapid and robust, Equipment-free or simple, and Deliverable to end-users) criteria for assessing POC diagnostics for sexually transmitted infections [22,49]. In order to effectively facilitate disease surveillance and disease control, especially in poor communities of the tropics, this became the standard for evaluation. Hence, all POC diagnostics must be ASSURED-compliant before they can receive approval and certification. This was strictly adopted by manufacturers of POC diagnostics, especially for detecting human viral infections including HIV, HBV, and HCV [23].

After a decade of implementing the ASSURED criteria, there was a unanimous call for a review to meet the technological advances of the current dispensation. The WHO/TDR approved the review and adopted the REASSURED (Real-time connectivity, Ease of specimen collection, Affordable, Sensitive, Specific, User-friendly, Rapid and robust, Equipment-free or simple, and Deliverable to end-users) criteria as the new benchmark for evaluating POC diagnostics [25]. The REASSURED evaluation process follows the guidelines below:

### 4.1. Real-Time Connectivity

This component of the REASSURED criteria assesses the availability of a reader for timely, reproducible, and smooth reading of the test results. This prevents variation in the interpretation of the test outcome and to allow convenient transmission of test results to the end-user [22,25]. In technological advancement, it has become necessary to connect and integrate POC tests into patients’ electronic medical records (EMR) such that test results will be stored and readily available in real time to the end-user [50]. One such system utilises the middleware electronic system, a data management system (DMS), to connect the IVD to the EMR, thus ensuring real-time access to the test results [50,51,52]. This promotes test results utilisation by clinicians. For example, in instances where a physician is out of reach or in decentralised settings, they could have access to POC test results through the hospital information system (HIS) for continuity of care [22]. Again, handwritten POC test results have been previously shown to be potentially limiting in the application of POC results as they may be non-readable and often presented with transcription errors [53,54]. According to Fung, 2020, manual entry increases the turn-around time, which defeats the rapidity and robustness associated with the POC test [50]. To reduce turn-around time, as well as avoid transcriptional errors that emanate from data management in POC tests, the use of either a uni-directional or bi-directional connectivity and DMS is highly recommended [50,55,56]. This system allows automated, wireless, and real-time transfer of POC test results to the end-users. Apart from POC results in linkage to the HIS and improving care, connectivity has proven to boost revenue collection by reducing financial losses associated with manual billing by at least 20%, as well as labour costs [57].

Globally, connectivity has been widely deployed in conventional laboratories of resource-rich and advanced settings for laboratory diagnosis by leveraging technology innovations and advancements [58,59,60]. This reduces the risk of errors and improves the integrity of laboratory results as observed with enzyme immunoassays (EIA), chemiluminescence immunoassays (CLIA), and nucleic acid testing (NAT) such as the RT-PCR for diagnosing HCV infections [61]. However, most POC IVDs are limited in the application of connectivity influenced by their intent of use, target population, and setting of use. For example, although the concept of HCV self-testing (HCVST) received strong acceptance in a global multinational study that sought to explore the values and preferences of targeted end-users of the test, respondents in Indonesia, Thailand, and Ukraine preferred a link of the test to a system where users could interact with clinicians by sending snapshots of their test results for assistance with reading and interpretation of results [13].

Commonly in LMICs of SSA, these devices are largely depended on by marginalised, hard-to-reach, and resource-limited settings, sometimes without access to a power supply and internet connectivity. For example, the OraQuick HCV POC test, Genedia, SD Bioline POC HCV test, the TriDot HCV test, Chembio HCV test, ARCHITECT HCV antibody test, Spot HCV, and Multiplo POC test, among others, have been widely deployed in similar environments in SSA [62,63,64].

However, the high demand for POC testing to promote equitable access to diagnostic services with linkage to care and treatment has prompted a shift in technological advancements and testing algorithms by the IVD industry to accommodate real-time connectivity in POC IVDs. For example, WHO recommends the Xpert^®^ HCV VL Fingerstick to produce a decentralised compact molecular test at POC that supports networking and connectivity, hence promoting real-time HCV surveillance in resource-limited settings [12,65,66]. The GeneXpert system has been widely deployed in SSA mainly by the National Tuberculosis Control Programs (NTCP) of various countries [67,68]. However, it is restrictively used and evaluated by a few countries for POC HCV RNA detection—for example, Malawi [69], Tanzania [70], and South Africa [71]. Similarly, the Genedrive POC HCV RNA testing system was introduced to provide a two-step real-time connected molecular confirmatory testing at POC [72]. Other CE-approved POC HCV RNA assays include the Truenat HCV RNA assay and the SAMBA II HCV Qualitative Whole Blood Test [73]. Although these innovative POC molecular closed systems may support connectivity, their application in hard-to-reach communities deprived of electricity may be questioned since these systems run on electricity. For instance, during an evaluation of the Xpert^®^ HCV viral load fingerstick assay in people who inject drugs in Tanzania, one sample failed to process due to a power cut during sample processing [70].

### 4.2. Ease of Specimen Collection

Central laboratories require invasive, hard-to-obtain, and sometimes painful procedures such as venepuncture, lumber tap, nasopharyngeal swabs, aspiration, and tissue biopsy, among others, to conduct a medical diagnostic test [69,74]. However, these procedures defeat the concept of POC, as all testing and results are not conducted at the site or near the patient. Ease of sample collection is one of the basic requirements for POC testing [22]. Easy-to-obtain, readily available, and non-invasive diagnostic tests have the potential to generate less discomfort for the patient, offer simpler and less risky testing procedures, and thus increase testing uptake and retention. This component influences the usability and acceptability of the IVD by the end-users [75,76].

Several IVDs for diagnosing HCV have been evaluated worldwide. These HCV IVDs were evaluated on varying populations and specimens—for example, dried blood spot (DBS) tests such as the Abbott ARCHITECT anti-HCV assay and the Xpert HCV viral load fingerstick [74,77,78,79]. The OraQuick and OraSure HCV POC tests run on oral mucosal transudate (OMT) or oral fluid [62,64,80,81,82,83]. The Chembio HCV Assay, SD Bioline HCV POC, advanced-quality one-step HCV test, Spot, Multoplo, and ARCHITECT HCV antibody test run on capillary whole blood [62,63,65,73,84,85,86]. Finally, the Advanced Quality Rapid Anti-HCV test, Immunocomb II HCV test, Genedrive HCV RNA, Acon HCV test, Assure HCV test, TriDot α, SD Bioline, and Genedia run on serum/plasma samples [3,62,64,72,81,84,87].

Among these, dried blood spot using easy-to-obtain capillary blood collected through a relatively painless and non-invasive finger prick and oral fluids are identified as alternative sampling procedures. This has been leveraged through technological advancements and novel innovations to provide simplified HCV testing systems and to improve test uptake [59]. Moreover, current HCV testing algorithms such as the Xpert^®^ HCV VL Fingerstick integrates microfluidic plasma separation without laboratory-interrupted conventional centrifugation [65,72]. This technology uses capillary whole blood from a relatively non-invasive finger prick to simplify the diagnostic pathway. This testing algorithm facilitates ease of specimen collection and creates convenience for end-users. In Australia, Belgium, and the United Kingdom, HCV RNA assays have been extensively evaluated and demonstrated to produce equally good clinical performance using serum/plasma and capillary blood; however, the latter was conveniently preferred to the former by the end-users [88,89,90,91]. Likewise, a non-invasive POC anti-HCV test using the OraQuick HCV rapid test in OMT and fingerstick capillary whole blood samples demonstrated high clinical performance in Spain [92].

In SSA, a prototype fingerstick whole-blood Triplex HIV/HCV/HBsAg self-test evaluated in the Democratic Republic of Congo received high acceptability by the end-users, on the basis of convenience and ease-of-sample collection among other features [85]. Moreover, the use of capillary whole blood (finger prick) by the Xpert^®^ HCV viral load fingerstick assay deployed to test PWIDs in Tanzania resolved the challenges of blood centrifugation and inconveniently invasive venous blood collected from the end-users, which sometimes require the service of expert phlebotomists [70]. For example, in the study by Mohammed et al., 2020, in Tanzania, they had challenges with venous sample collection, and hence they failed to obtain venous samples from 4% of the study participants, with another 1% attributed to inadequate sample volume [70]. Self-testing for HCV antibodies in general populations of Egypt demonstrated high usability and acceptability of using the non-invasive oral fluid OraQuick^®^ HCV Rapid Antibody test [75]. Similarly, in South Africa, oral fluid, capillary whole blood, and DBS demonstrated sufficient performance and is recommended as alternative samples for the OraQuick HCV POC test and the ARCHITECT HCV antibody assay [63]. In Ghana, there is limited information on the use of non-invasive methods to detect HCV. However, there is evidence of using invasive blood collection methods such as venepuncture for both rapid HCV and NAT testing [4,93,94,95,96].

### 4.3. Affordability

The WHO aims to prevent approximately 2.1 million HCV deaths and 10 million new HCV infections by 2030 [9,97]. Scaling-up testing is a pre-requisite to achieving these goals, among others, on the itinerary such as strategic prevention campaigns and treatment [98]. However, there is limited funding and financial commitment to effectively diagnose all 45–85% of global undiagnosed HCV cases with linkage to care and treatment [99]. There is no dedicated global funding support for HCV, unlike other diseases such as HIV, HBV, TB, and malaria, among others [100]. The global HCV elimination goal has a funding gap of about USD 41.5 billion, hence requiring innovative and alternative financing approaches to access funds from the global community [98,101]. For instance, the in-country reallocation of existing funds towards HCV elimination from within existing donor programs such as the Global Fund’s fight against AIDS, Tuberculosis, and Malaria (GFATM), and moreover, support from the pharmaceutical and diagnostic industry, performance-based financing, donor-support from high-income countries, and public–private partnerships, among others, may provide alternative funds [102].

The cost of testing for HCV has a toll on the elimination prowess, especially for largely donor-funded LMICs [18,98,100]. This is because the cost of logistics for diagnostic testing has a direct link to accessibility and affordability in LMICs [103]. Many underserved groups in LMICs call for HCV testing, especially HCVST, to be free [13]. In the IVD market, the OraQuick HCV costs GBP 12–15 per single-use test [104]. The HCV EIA serological assay costs USD 1–9 per test, HCV RDTs cost USD 0.5–7, and laboratory-based NAT costs between USD 30 and 120 [59]. On the other hand, NAT analysers that can detect HCV RNA at POC range from USD 10,000 to 25,000, excluding fixed costs [100]. The reagents for this product range from USD $14 to 30 per test. Furthermore, laboratory-stationed analysers for HCV RNA cost as low as USD 100,000, and consumables cost from USD 9–50 per test [100]. The WHO therefore re-strategised to decentralise confirmatory testing to hard-to-reach and poor communities that cannot afford the expensive laboratory-led testing. The POC HCV RNA testing using the Xpert^®^ HCV viral load fingerstick assay is recommended in these settings to boost test uptake towards HCV elimination in 2030 [12]. Generally, the WHO HCV testing guidelines require a two-step testing initially with antibody testing at POC followed by a confirmatory test with HCV RNA assays [9].

Globally, several countries have assessed and evaluated cost-effective approaches to saving costs whilst aiming to increase HCV diagnosis and treatment services. In Georgia, it costs on average USD 1 for an antibody RDT test per test, USD 15 per test for an HCV-RNA on-site test (Genexpert), and USD 40 per test for an HCV-RNA test referred to a central laboratory [103]. However, an evaluation report of a new testing pathway saved them costs. Thus, the testing pathway using an on-site HCV-antibody rapid test followed by HCV RNA testing, an on-site Fibroscan, and treatment costs USD 139 per case [103]. This pathway saved the country USD 127,052 per 10,000 cases tested compared with their standard testing and treatment pathway of on-site rapid testing, patient travel for HCV RNA test confirmation, patient travel for two Fibroscan tests, and patient travel for HCV RNA testing for monitoring during treatment. In an emergency department in the UK, an HCV antibody test cost of GBP 3.64 per test and HCV RNA test cost of GBP 68.38 per test were demonstrated to be highly cost effective with GBP 8019 per quality-adjusted life-year (QALY) gained [105]. Similarly, HCV antibody testing costs between USD 3.14 and 20.0, whilst an HCV quantitative RNA test costs between USD 31.70 and 150.0 in South Korea [106]. Moreover, in the USA, the HCV antibody test, qualitative NAT, and quantitative NAT averagely cost USD26.31, USD 64.70, and USD 78.97, respectively [107]. These unit costs for HCV testing may seem more expensive in LMICs and hence may cause low uptake in HCV services. However, they demonstrate high cost-effectiveness in higher-income countries.

In SSA, on average, the HCV-RNA POC test costs EUR 13.68 in Cameroon, Côte d’Ivoire, and Senegal each, whereas laboratory-led conventional HCV-RNA testing costs EUR 95.30, EUR 45.70, and EUR 68.60, respectively [18]. A decision analysis model in a multinational evaluation of the cost effectiveness of HCV self-testing in China, Georgia, Vietnam, and Kenya revealed a higher cost (USD 361) of HCV diagnosis with EIA in Kenya [108]. However, the addition of a POC HCV self-testing platform as a first-step test followed by confirmation with EIA further increased the cost to USD 587. Comparatively, in the absence of HCV self-testing, the cost of diagnosing HCV using the standard EIA test in Vietnam, Georgia, and China are averagely USD 35, USD 55, and USD162, respectively. However, the addition of a self-test to their standard of testing using EIA increased the cost to USD 104 in Vietnam, USD 163 in Georgia, and USD 2647 in China.

Egypt embarked on a comprehensive program in 2014 under the national strategy to eliminate HCV as a public health threat by 2021 [109]. This strategy included public campaigns, free screening and testing, free treatment, and follow-up visits to ensure medication compliance and reduce stigma. This strategy demanded huge financial commitment. According to the World Bank, as of 2015, Egypt’s HCV economic burden stood at USD 3.81 billion, representing 1.4% of the country’s GDP [109,110,111]. The Egyptian government spent as low as USD 0.60 per kit on purchasing the HCV screening kits (Abbott SD Bioline HCV) and USD 4.80 per test on confirmatory test using the Cobas AmpliPrep/CobasTaqMan HCV Test, Roche Diagnostics quantitative real-time PCR. The World Bank revealed a USD 530 million estimated cost for Egypt’s HCV elimination drive [111]. However, Egypt’s ambitious strategy toward eliminating HCV transcends to other African countries by supporting 1 million people in 14 African countries with HCV testing and treatment [112].

Several attempts have been made to provide access to cheaper HCV diagnostics, especially in poor countries in SSA. For example, in an effort to provide equal or better diagnostic performance for HCV diagnosis in poor settings, Jülicher and Galli, 2017, evaluated an alternative algorithm for confirming HCV infection at a lower cost [113]. They discovered that replacing the laboratory-led HCV RNA with HCV antigen in a confirmatory test for a positive HCV POC antibody test significantly reduced total costs by USD 2.74 per screening test and costs per diagnosed infection by USD 44. However, this approach detected fewer active infections in a deficit of 110 per 100,000 individuals screened. Further confirmatory tests using the laboratory-led HCV RNA only reduce the cost by USD 1.16 per subject screened and USD 22 per case detected compared to using the laboratory-led HCV RNA as a confirmatory test for the HCV POC antibody test. On the other hand, though using the HCV antigen test as a screening test to detect active infections recorded the highest infection rate, this approach had the highest cost implication at +USD 3.80 per subject, +USD 323 per case detected vs. standard. Again, the recommendation of the POC HCV RNA assay as a more convenient and cheaper alternative to the expensive and centralised laboratory-led HCV RNA confirmatory testing will give value for money whilst increasing test uptake in LMICs [12].

Multiplexing, as described as the future of POC testing, demonstrates a cost-effective and cheaper diagnostic algorithm for HCV testing [22,59,114,115]. Integrating several diseases in a panel of tests increases uptake whilst strengthening multiple diseases and co-infection surveillance [59].

### 4.4. Sensitivity and Specificity

The clinical performance (sensitivity and specificity) of IVDs is a key component in assessing the accuracy and utility of diagnostic tests [116,117]. These parameters measure the ability of a diagnostic test to adequately measure the outcome of interest (disease) and to give definitive information about the presence or absence of the disease [118,119]. The WHO PQ process recommends a sensitivity of ≥98% and a specificity of ≥97% as the required accurate performance criteria for rapid IVDs [120]. However, in-country-specific performance evaluation are also recommended to establish populations-specific performance.

Globally, different POC HCV IVDs have demonstrated varying clinical performances compared with the reference laboratory testing available in that setting. For example, the OraQuick HCV POC using enzyme immunoassay (EIA) as the gold standard recorded a sensitivity of 94.6% (95% CI 90.0–99.2%) and a specificity of 100% in Estonia [82]. However, in the USA, the OraQuick HCV test significantly established the highest accurate performance with a sensitivity of 99.4% (95% CI 98.0–99.9%, *p* < 0.05) and specificity of 99.7% (95% CI, 98.6–100%) among the Instant, Axiom, CORE HCV, and FirstVue HCV tests) [121]. In Korea, the SD Bioline HCV test demonstrated higher clinical performance (sensitivity: 78.8% (95% CI: 71.2–86.8%) and specificity: 100%) than the Genedia rapid test (sensitivity and specificity of 69.7% (95% CI: 61.1–78.9%) and 99.3% (95% CI: 97.0–100%), respectively) using the recombinant immunoblot assay (RIBA) as a reference test [122]. In Germany, the rapid onsite anti-HCV used as a screening tool established 99% sensitivity and 88% specificity using the Architect anti-HCV test as the gold standard [123]. In addition, both Bioeasy^®^ and Immuno-Rapido HCV^®^ POC HCV tests provided an equal performance with a sensitivity of 97.1% and specificity of 100% when compared with CLIA in Brazil [124].

Arguably, the majority of the IVDs are designed and clinical trials are conducted, especially on accurate performance, in non-African populations before product exportation. It is therefore appropriate for IVDs to be effectively evaluated and approved on the populations they are meant for. In LMICs in SSA, especially in hard-to-reach settings, much emphasis is placed on accurate performances of POC tests, unlike in high-income countries that have better alternatives and access to laboratory-confirmed HCV testing. It is therefore important for these tests to be evaluated in African indigenous populations where the IVDs are intended to be used. A few of these POC IVDs have been evaluated in Africa. In Cameroon, the Hexagon HCV rapid test using the HCV-RNA quantitation with Cobas TaqMan HCV as reference tests gave a sensitivity and specificity of 70.3% (95% CI: 55.6–85.0%) and 99.4% (95% CI: 99.1–99.7%), respectively [125]. In Egypt, the One-Step ImmunoComb II HCV test, the ACON HCV rapid test, and HCV TRI-DOT rapid tests were evaluated on accurate performance using the Dialab HCV Ab ELISA kit as a reference test [126]. All three tests demonstrated high test accuracy (the ACON HCV rapid test sensitivity: 98% (49 of 50), specificity: 100% (50 of 500); the HCV TRI-DOT rapid test sensitivity: 98% (49 of 50), specificity: 100% (50 of 50); and the ImmunoComb II HCV test sensitivity: 96% (48 of 50), specificity: 100% (50 of 50)). Moreover, in South Africa, the OraQuick HCV POC test and the ARCHITECT HCV antibody tests were evaluated in terms of performance using the HCV viral load (COBAS Ampliprep/COBAS TaqMan version 2) assay as a reference [63]. The OraQuick HCV POC test recorded 98.5% (95% CI 97.4–99.5) sensitivity and 98.2% (95% CI 98.8–100) specificity, whereas the ARCHITECT HCV antibody test showed a sensitivity of 96.0% (95% CI 93.4–98.6) and specificity of 97% (95% CI 94.8–99.3).

On the other hand, the POC HCV RNA molecular assays, and thus the Xpert HCV VL Fingerstick assays, the Genedrive HCV ID Kit, the Truenat HCV RNA assay, and the SAMBA II HCV Qualitative tests, demonstrated high performance, irrespective of the setting and population, including Africa with a pooled sensitivity and specificity of 99% (95% CI: 98–99%) and 99% (95% CI: 99–100%), respectively [73].

### 4.5. User Friendliness

This REASSURED component evaluates how easily a diagnostic test can be operated or used by the intended end-users with little or no training [25]. It is a key predictor of the usability and acceptability of IVDs by the end-users, especially during self-testing [75,76]. Ideally, POC testing is intended to provide access to diagnostic tests in non-laboratory settings, and hence they are used by non-laboratory-trained professionals such as nurses, clinicians, midwives, and sometimes patients during self-testing [127]. These intended end-users have no knowledge and expertise in laboratory testing and the use of IVDs. These have raised issues regarding quality assurance discrepancies and bottlenecks associated with quality controls, troubleshooting, documentation, and professional competencies [128]. However, with the widespread demand for POC testing, especially in hard-to-reach and resource-limited settings with no access to central laboratories, this may be addressed with continuous refresher training and competency assessment for the end-users [129,130,131]. Refresher training seeks to enhance compliance with quality assurance practices to provide equitable and continuous diagnostic services to marginalised populations.

A user-friendly diagnostic test has evolved to include an instruction for a user guide or quick reference guide (written insert or digital interphase). These instructions are presented in multiple languages with clear instructions on the test protocol; test principle; quality control; test result interpretation; troubleshooting; storage, and usage temperature requirement; and cleaning and disposal requirements, among others [86,132]. This promotes uptake and convenience of use [133]. For example, the emergence of the COVID-19 pandemic caused the main shift in where diagnostic products are used, especially from laboratories to homes [134,135]. Similarly, to enhance the convenient uptake/usability and acceptability of HCV IVDs as POC tests by end-users, they must meet this criterion of user friendliness.

Studies have evaluated the user-friendliness of POC HCV diagnostics globally. However, a majority have been accepted and documented as easy-to-use tests on the basis of their possibility of use in self-testing. For example, the OraQuick^®^ HCV Rapid Antibody Test was used in a pilot study for men who have sex with men (MSM) in China [136]. Vietnam is considering the application of HCVST to augment existing HCV diagnostic services to boost coverage among high-risk populations beginning with the OraQuick^®^ HCV Self-Test (prototype) [76]. However, the users of the OraQuick^®^ HCV Self-Test (prototype) encountered difficulties in reading and understanding the test manual/insert, opening the test pouch, removing test tube caps, sliding the test tubes into the test stand, placing the test kit into the test tube, and interpreting the test report [76]. These difficulties called for clinical staff support and assistance for optimum test performance.

A few of the HCV POC tests have been evaluated in Africa among target populations by non-laboratory users, especially HCVST. For example, in South Africa, three late-stage HCVST prototypes (SD Bioline HCV, Care Start, and First Response HCV) received high acceptability, usability, and ease-of-use by lay end-users with few errors emanating mostly from improper handwashing, inaccurate sample collection and transfer practices, and interpretation of invalid results by the end-users [137]. Similarly, in Egypt, OraQuick HCVST users struggled to open the test package, remove the cap of the test tube, place the tube on a stand, place the test kit into the tube, and read/interpret the test results [75]. However, about 12.1% of the end-users received a form of assistance in using the test.

Although several rapid IVDs tests have met the WHO PQ process, there is limited information on their evaluation of their user-friendliness in the primary healthcare clinic (PHC) settings by non-laboratory staff in LMICs, specifically in SSA. This may be attributed to barriers to accessing the tests such as poor procurement practices and lack of training and financial support for non-laboratory staff (doctors and nurses) in PHC clinics [138,139].

### 4.6. Robustness/Rapidity

As determined by laboratories, the turnaround time (TAT) of an IVD test is largely influenced by the time difference between when specimens are received in the laboratory and the time results reach the patient [140]. This means that a delay at any stage of the diagnostic process affects the dispatch of the laboratory test. TAT may vary in laboratory settings and PHC POC test sites due to pre-analytic and post-analytic quality processes and discrepancies. However, the time it takes IVDs to produce results (the analytical phase) is the same worldwide, as determined by the manufacturer [65,141,142]. For example, in clinical laboratories in the USA, it takes between a few days to a few weeks for HCV specimen to be collected, tested and results dispatched; however, it takes 20 to 30 min at the PHC clinics to obtain HCV test results using the POC rapid anti-HCV IVD [143]. The observed longer TAT for laboratory-led conventional HCV testing is influenced by the longer duration of sample collection (venous blood), sample transportation to a central laboratory, sample processing, validation, and dispatch of results.

A robust POC test algorithm must meet at least a 30 min turn-around time, and the test result must reach the patient within 30 min of testing [16,116,144]. Several HCV IVD tests meet this classification. For example, the OraQuick HCV test is evaluated globally as a POC test and HCVST. The OraQuick HCV test produces diagnostic results on average by 30 min using varying required samples such as whole blood, oral fluid, serum, and plasma [80,104]. Others such as the Advanced Quality One Step HCV Test provides results in 6 min, the HCV Bi-Dot in 3 min, the HCV Rapid test Bioeasy in 15–20 min, the Chembio DPP HCV test in 15–30 min, the Genedia HCV Rapid LF in 20–30 min, the Hexagon HCV in 5–20 min, the ImmunoComb II HCV test in 10–15 min, the Multiplo Rapid HIV/HCV Antibody Test in 3 min, the One Step HCV Rapid Test in 10 min, the Toyo anti-HCV test in 5–15 min, the Anti-HCV Antibody Rapid test in 3 min, the SD Bioline HCV test in 5–20 min, the SeroCard HCV in 19 min, the SM-HCV Rapid Test in 3 min, the GLD HCV-SPOT in 10 min, and the HCV TriDot Rapid in 9 min [62]. These rapid POC HCV IVDs have been evaluated globally and accepted to produce rapid HCV test results for real-time linkage to care and treatment specifically in hard-to-reach settings of LMICs.

However, unlike the conventional HCV RNA test, the Xpert^®^ HCV VL Fingerstick produces same-day on-site HCV RNA results in 60 min [65]. This molecular closed system was deployed and recommended by the WHO as an alternative for conventional laboratory test that takes days to produce HCV RNA confirmatory results in resource-limited settings and LMICs. This promotes real-time accessibility to diagnostic results to link them to care and treatment. Similarly, the Genedrive HCV ID Kit, the Truenat HCV RNA assay, and the SAMBA II HCV POC qualitative tests produce HCV RNA results within 2 h to reliably inform medical decision making in real time [73].

### 4.7. Equipment Free or Simple

The traditional diagnostic techniques for diagnosing HCV require laboratory-intervened, centralised, advanced, or complex laboratory systems powered by electricity (RT-PCT, EIA, and CLIA among others) [74,145]. However, this system continues to discriminate against and deprive high-risk target populations (MSM, PLHIV, PWID/UD, prison inmates, pregnant women, and blood donors) in hard-to-reach and resource-limited settings. This contributes to HCV underdiagnoses in about 80% of the target populations and hence is a critical setback to WHO HCV elimination 2030 [99]. Due to complexities in the laboratory machinery, over-dependence on electricity, and inadequate laboratory infrastructure in LMICs, there is a main shift to POC IVDs that tend to serve the needs of underserved settings. This component of the REASSURED criteria therefore assesses how simple and portable the POC diagnostic is compared to instrument or large-equipment-dependent diagnostics in the laboratory space.

The WHO guidelines on viral hepatitis testing provide the basis and rationale for simplification of testing algorithms to increase affordability, acceptability, and uptake [12,144]. Globally, several POC HCV IVDs agree with this criterion—for example, the OraQuick HCV POC, Genedia, SDBioline POC HCV, the TriDot HCV, Chembio HCV, ARCHITECT HCV antibody, Spot HCV, and the Multiplo HCV tests [62,63,64]. These POC IVDs operate on lateral flow immunoassay (LFA) configured in a portable test cassette or strip [146]. They require capillary whole-blood specimens easily and conveniently obtained through a finger prick or oral fluid obtained through a swab and are devoid of complex sample management processes such as conventional centrifugation.

However, the new simplified POC HCV RNA testing algorithms deviate from this criterion. For example, the Xpert^®^ HCV VL Fingerstick, the Genedrive HCV ID Kit, the Truenat HCV RNA assay, and the SAMBA II HCV POC operate on a stationed automated equipment-based molecular system [73].

### 4.8. Deliverable to End-Users

There are end-user discussions on barriers to geographical access and stock-outs to HCV POC tests in resource-limited settings [13,147,148]. The WHO 2020 global report sought to accelerate access to HCV diagnostics with linkage to care and treatments as a step to overcome barriers in LMICs [148].

Accessibility to HCV diagnostic services is a global challenge that may be attributed to a lack of funds and commitment, unlike other infectious diseases such as HIV, TB, hepatitis B, and COVID-19 [98,100,101]. The WHO recommends HCV POC testing (self-testing) as a valuable tool to increase HCV awareness, voluntary testing uptake, and testing coverage, especially in HCV endemic settings and hard-to-reach target populations [13]. In a multinational exploratory study on the values and preferences of HCVST end-users, there were challenges with continuous access and deliverability [13]. Moreover, in India, a cross-sectional study reported low access to HCV diagnostic services among PWI/UDs [147]. Of the 5777 HCV antibody-positive persons recorded, only 5.5% knew of their HCV status, 3% had seen a physician, and more than 50% had never had an HCV test since they did not know any of such tests, whilst 14.3% did not know where to access the test. In a 10-year review in the USA in 2014, out of the estimated 3.5 million people with chronic HCV infection, only 50% (95% CI, 43–57) were clinically diagnosed and were aware of their status, 27% (95% CI, 27–28) had laboratory confirmation with HCV RNA testing, and 43% (95% CI, 40–47) could access care in PHC clinics [149]. However, in 2020, there was improved accessibility and deliverability to HCV awareness and diagnostic and treatment services with the influx of non-invasive tests at POC centres (community health centres) [150].

A survey in some LMICs including SSA revealed limited HCV testing in community-based and PHC clinics [100]. Most of these clinics run a patient co-payment policy due to a lack of sustained funding commitments [100]. Most of the diagnostic services were centralised in the hospital settings denying access to the hard-to-reach settings. There are recommendations for the provision of HCV testing at the point of sales such as in pharmacies, PHC clinics, and mobile vans to facilitate easy access to community testing, especially in hard-to-reach, secluded, and poor settings [14]. Others include the need for governments of LMICs to demonstrate political will through funding commitments and public partnerships to ensure continuous accessibility and deliverability of HCV diagnostic services in PHC clinics [151].

## 5. Implications of the REASSURED Criteria for Practice and for WHO HCV Elimination Goal 2030

Globally, there are sizeable hard-to-reach and resource-limited communities in LMICs that may be off the national electricity grid, especially in SSA. These communities lack access to simple, robust, affordable, and user-friendly HCV diagnostic services. There is the need for health systems to select and procure HCV IVDs that can serve intended purposes at POC. This helps to effectively meet the WHO strategic goal to eliminate viral hepatitis as a public health problem by reducing new viral hepatitis infections by 90% and viral hepatitis deaths by 65% by 2030 [9]. However, an effective POC diagnostic must meet the required analytical and clinical validation stipulated in the REASSURED criteria.

Over a decade, the WHO ASSURED criteria had out-lived its purpose of serving the populations with effective and quality IVDs [23]. Although largely successful, the advent of the innovative technology systems and platforms necessitate the need to consider integrating into IVDs and diagnostic service delivery the rapid evolution of how potential end-users of these test devices and services communicate and interact. Ultimately, the REASSURED evaluation of HCV POC testing better facilitates, in real-time, the concept of universal health coverage and primary health care as part of Sustainable Development Goal 3 [25,152].

Using a rapid and connected POC HCV IVD, test results must reach the clinicians, prescribers, and patients in real time to enhance linkage to care and effective treatment. This promotes early diagnosis of acute infections in high-risk groups that prevents ongoing chronic disease progression. Early detection saves about 80% of asymptomatic acute HCV infections from progressing to chronic infection [9]. In addition, real-time connectivity of HCV IVDs in HCVST provides a convenient platform for target populations in hard-to-reach and poor settings to access result interpretation, care, and treatment in real time, irrespective of their geographical location [13]. In this review, although all the POC HCV IVDs met the rapidity criteria of producing results within 30 min, connectivity was a challenge. However, the WHO-recommended POC HCV RNA confirmatory tests complied with real-time connectivity [73].

Ideal POC IVDs must meet the criteria of using readily available and unprocessed specimens that require laboratory intervention [22]. The use of unprocessed and conveniently obtained specimens such as capillary whole blood and oral fluids for POC HCV IVDs meet the REASSURED criteria of ease of specimen collection. Moreover, the ease of use and user-friendliness of POC HCV tests, especially in HCVST, for example, the inclusion of an easy-to-read test manual and guidelines, enhances the acceptability and usability of the IVD [75,76]. Improved HCV test uptake is a critical requirement for meeting the HCV elimination target 2030.

Expensive HCV testing is a concern and a barrier in LMICs, particularly in SSA, since it has a toll on the HCV elimination program [18,98,100]. Preferably, an HCV elimination program should be run freely from financial constraints for target groups. For example, Egypt committed to an ambitious HCV elimination plan in line with the WHO HCV elimination target to end HCV in Egypt by 2021 [109]. This received a strong HCV test uptake largely because of the free testing incentive attached. However, the HCV elimination target 2030 by the WHO has met huge funding gaps that need innovative and alternative financing routes [101,102]. The IVD industry has a role to play in achieving this goal by producing cheaper but quality alternative HCV IVDs that meet LMICs’ target populations as well as budgets of governments and their partners.

POC HCV IVDs of high accurate performance are critical to HCV elimination [153]. Largely, the goals of IVD manufacturers and standard regulatory bodies are to produce and approve high-performing (sensitivity and specificity) IVDs, respectively [27,42,120,145]. Clinical performance of POC HCV IVDs may not necessarily result in diagnostic discrepancies in high-income countries due to the availability and accessibility of laboratory-led advanced HCV testing. However, the evaluation of POC HCV IVDs and its associated challenges in LMICs, particularly SSA, cannot be overlooked. The sub-region largely depends on POC HCV IVDs for diagnostic services due to the unavailability and inaccessibility of decentralised laboratory services, especially in hard-to-reach and resource-limited settings [14,39,112,154,155]. In this review, varying clinical performances of POV HCV IVDs were established in varying target populations. However, there were limited information of test accuracy in SSA populations.

The hospital-based centralised laboratory system in LMICs, particularly SSA, provides restricted HCV diagnostic services to only a section of the population and thus populations within the catchment areas. This creates a sizeable underserved, marginalised, and neglected HCV target groups in hard-to-reach and poor communities. This calls for the need for the industry to produce equipment-free, simple, and portable IVDs that provide continuous HCV deliverable services to these underserved populations. In the current review, most of the POC anti-HCV IVDs met the criteria of simple and equipment-free IVDs. However, the confirmatory testing with POC HCV RNA deviated from this criterion [73]. Moreover, there was limited deliverability of these tests to the target populations largely due to funding gaps, lack of political will, and poor government commitments [151].

## 6. Recommendations for Future Research

This review found that most of the studies largely rely on the use of laboratory-based clinical performance to evaluate the validity of the HCV IVDs. Moreover, these performance evaluations were carried out in high-income countries and on non-African populations. We recommend that future studies be conducted to also ascertain the usability and acceptability of the POC HCV IVDs in non-laboratory settings and among populations where the IVDs are intended to be used particularly SSA. Furthermore, ease-of-use and usability of the POC HCV tests were commonly evaluated using HCVST. We recommend future studies to explore the evaluation using non-laboratory end-users in PHC clinics who offer HCV services. We also found that there is limited information on the deliverability of POC HCV testing in hard-to-reach and resource-limited settings of SSA. We recommend future studies to assess the deliverability of POC HCV testing in SSA. Finally, there was limited information on the cost-effective assessment of HCV IVDs in LMICs particularly in SSA. We recommend that future studies be conducted to evaluate the cost effectiveness of using POC HCV IVDs in SSA.

## 7. Conclusions

In summary, movement of IVD products in SSA is restricted due to lack of continent-wide standards and regulatory authority. The WHO PQ process and the REASSURED criteria set the product strengthening and validation guidelines in SSA as adopted by country-specific standards and regulatory agencies. This review presents evidence of regulatory bodies and industry’s over-reliance on the use of clinical performance in evaluating HCV IVDs. In doing so, the other critical components of the criteria that seek to establish the usability and acceptability of the IVDs at POC are often neglected. Every component of the REASSURED criteria is equally important to advance the goal of HCV elimination. In addition, this review reveals the dearth of research on the evaluation of the POC HCV IVDs in non-laboratory settings of LMICs, particularly in SSA. Finally, this review sets the tone for a multinational study to evaluate the Bioline HCV POC testing in SSA.

## Figures and Tables

**Table 1 diagnostics-13-00684-t001:** Standards for clinical diagnostics.

Context	Regulator	Standards and Criteria
Global standards	World Health Organization (WHO)	ASSURED criteria REASSURED criteria
	Global Harmonization Task Force (GHTF) and International Medical Device Regulators Forum (IMDRF)	GHTF/SG1/N78:2012, GHTF/SG1/N77:2012, SG1 N071:2012, N044:2008, GHTF/SG3/N19:2012
	International Organization for Standardization (ISO)	ISO/AWI 15193/4, ISO6717:2021, ISO21151:2020, ISO17822:2020, ISO18113-1:2009, ISO22870:2016
	The Acute Care Testing organisation	Five-stage pocket checklist for evaluating POC diagnostics
Continental standards	Asian Harmonization Working Party (AHWP)	ISO/TC 212, ISO/TC 210
	European Union (EU)	IVDD (Directive 98/79/EC) IVDR (Regulation (EU) 2017/746)
Sub-Saharan Africa standards	In-country Food and Drug Authority (FDA) and Standards Authority	WHO Prequalification process (WHO ASSURED/REASSURED)/ISO criteria

## Data Availability

Not applicable.

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
