# Peer review of "Quality Assurance for Hepatitis C Virus Point-of-Care Diagnostics in Sub-Saharan Africa"

_diagnostics, 2023, doi:10.3390/diagnostics13040684_

Round 1

Reviewer 1 Report

A very well written and comprehensive assessment of the state of play for HCV testing around the world, but particularly in SSA. I believe it will be a good resource to many working in HCV elimination. Pls see my comments to the editor when they are released.

Reviewer 2 Report

The manuscript entitled "Quality Assurance for Hepatitis C Virus Point-of-Care Diagnostics in sub-Saharan Africa" by Duah et al. is an interesting and comprehensive review of regulatory standards and quality indicators for validation and approval of clinical diagnostics on the topic of HCV testing for high-risk populations, particularly in sub-Saharan Africa.

The review is very well written and well structured and provides, to my knowledge, the necessary evidence regarding the above topic.

From my perspective, there are no major concerns. In my opinion, the manuscript is suitable for publication in Diagnostics.
